# Suppression of *Ganoderma boninense* Using Benzoic Acid: Impact on Cellular Ultrastructure and Anatomical Changes in Oil Palm Wood

**Rozi Fernanda** [1] , **Yasmeen Siddiqui** [1,*] , **Daarshini Ganapathy** [1] , **Khairulmazmi Ahmad** [1,2,*] and **Arthy Surendran** [3]

1 Laboratory of Sustainable Agronomy and Crop Protection, Institute of Plantation Studies, Universiti Putra Malaysia, Serdang 43400, Malaysia; rozifernanda86@gmail.com (R.F.); daarshinig@gmail.com (D.G.)

2 Department of Plant Protection, Faculty of Agriculture, Universiti Putra Malaysia, Serdang 43400, Malaysia

3 School of Life Sciences, University of Warwick Wellesbourne, Warwick CV 359EF, UK; Arthy.Surendran@warwick.ac.uk

* Correspondence: yasmeen@upm.edu.my (Y.S.); khairulmazmi@upm.edu.my (K.A.)

**Abstract:** Basal stem rot (BSR) caused by a wood degrading fungus, *Ganoderma boninense*, is the major constraint in palm oil production. It degrades the wood components and causes palms to collapse, leading to heavy losses. Inefficacy in controlling this disease could be caused by the lack of understanding in how the pathogen establishes itself on the host concerning wood decay stages. This study aimed to understand and determine the role of benzoic acid on the suppression of *G. boninense* and production of ligninolytic enzymes responsible for wood decay. Further, the alteration in wood component structure due to *G. boninense* infection and its prevention were studied. Benzoic acid treatment resulted in more than 80% of inhibition in *G. boninense* growth. SEM and HR-TEM analysis confirmed the antifungal activity of benzoic acid by disruption of mycelial morphology and cellular ultrastructure. Moreover, the membrane permeability assay recorded enhanced cell mortality in benzoic acid treated mycelium. The degradation of oil palm woodblock caused 58.86 % wood dry weight loss at day 120. In contrast, reduction in dry weight loss (58.82%) was recorded in woodblock treated with concentrations of benzoic acid of 5 mM and above. It is concluded that the use of benzoic acid could inhibit or delay pathogen establishment in oil palm wood, leading to the sustainable management of BSR disease. Further, glasshouse and field trials are required to prove the consistency in current findings which may contribute to reduced land expansion to create new disease-free land for oil palm planting.

**Keywords:** *Ganoderma boninense*; basal stem rot; flowcytometry; membrane integrity; wood degradation; ligninolytic enzyme; cell wall degradation; benzoic acid

## 1. Introduction

Oil palm (*Elaeis guineensis Jaq.*) is one of the most extraordinary oil crops that contributes to the economic development of its producing country. Indonesia and Malaysia dominate the production of palm oil, contributing 84.87% of total production in 2019 [1]. In Malaysia, oil palm was the major contributor to the gross domestic product (GDP) of the agriculture sector in 2018 at 37.9% compared to other crops [2]. To fulfill global demands, the production of this oil is increasing annually by land expansion for oil palm cultivation [3]. Malaysia's land expansion for palm oil has been from 5.74 to 5.81 million hectares from 2016 to 2018 according to the Malaysian Palm Oil Board (MPOB) [4]. Further, land expansion has been controlled by the Malaysian government by keeping 50% of its forest cover intact [5]. However, the area covered by smallholder plantations in Indonesia increased every year and in 2018 smallholder plantation covered 5.8 million hectares, representing nearly half (40.62%) of the Indonesian oil palm plantations [6]. Currently, oil

palm plantations cover 14.9 million hectares of land in Indonesia [7]. A global expansion of 43% in oil palm cultivation of 15 million has been reported [8] endangering forest reserves.

World palm oil production yields in Malaysia and Indonesia have increased, especially between 1998–2008, when yields increased 4% annually. However, an unexpected decline in growth pattern has been observed since 2009 [9], indicating that significant events have occurred. A factor in the decline may be that oil palm is prone to attack by various diseases [10]. The most severe disease that has been identified in oil palm is Basal Stem Rot (BSR) caused by a wood degrading fungus, *Ganoderma boninense* [11–13]. This fungus is a saprophyte and can persist for a long time in debris, stump and the leftover roots of unattended logs in the planation [14]. It has been suggested that the disease incidence (DI) is cumulative in successive replantation. Susanto et al. (2005) reported that in the first replantation, oil palm plants of 10 years old have been affected with less than 2% of DI; however, at the third replantation DI (70%) has alarmingly increased [15]. Therefore, new lands are used for the plantation, encouraging deforestation. This fungus has brought a significant loss in palm oil production by reducing the yield and killing the oil palm tree. Assis et al. (2016) reported economic losses of 43.32% of the potential yield in six months [16]. Similarly, the fruit bunch yield losses due to the BSR in oil palm between 0.04 and 4.34 t ha$^{-1}$ from 10 to 22 years of planting has been reported and it is predicted that more than 60 million mature oil palms could be infected in Malaysia [17–19]. The infected palms with external symptoms having over five fruiting bodies (Basidiocarp) of *G. boninense* did not produce any fruit bunches, and 46.8% of infected palms were dead or collapsed twelve months after the external symptoms were identified [20]. The presence of basidiocarps shows that the fungus has successfully been established in the wood for at least several years, and extensive degradation could have taken place in the oil palm trunks [21].

*Ganoderma boninense* is a white-rot fungus due to its ability to enzymatically degrade lignin, a component of oil palm wood. This enzymatic degradation causes a decrease in wood strength during the initial stages of decay [13,22,23]. White-rot fungi play a crucial role in the short-term conversion and degradation of wood polysaccharides [13,24]. It can degrade the wood components of the plants by either simultaneous degradation or selective degradation. Simultaneous degradation breaks down all structural components, lignin, cellulose and hemicellulose, whereas selective degradation breaks down lignin and hemicelluloses, leaving cellulose-rich wood [13,25–27]. The degradation of these components results in the rapid reduction of wood weight [13,23].

The degradation of oil palm wood by *G. boninense* starts when mycelia penetrate the plant cell through an unwounded root and colonize the lower stem of the palm [27]. Successful penetration and further degradation of oil palm wood by *G. boninense* depend on the production of ligninolytic and hydrolytic enzymes [28,29]. Ligninolytic enzymes are the key to the penetration process through oil palm roots and involvement in the degradation of lignin. These enzymes include two peroxidase enzymes, which are lignin peroxidase (LiP) and manganese peroxidase (MnP) and laccase enzymes [29,30]. The production of these enzymes is essential for *G. boninense* to penetrate and overcome the first barrier of plant defence, and degradation of lignin is considered to be the rate-limiting step in the infection process [29].

The current control measures for BSR management including cultural practices, mechanical and chemical treatment have not proved satisfactory. Alternative control measures to overcome the *G. boninense* problem are focused on the use of biological control agents and many potential bioagents were identified with little proven practical application [31]. Inefficacy of control of this devastating disease could be due to the lack of understanding of how a pathogen establishes itself on the host and the role of *Ganoderma* infection concerning wood decay stages in oil palm wood. In a recent study, the use of plant secondary metabolites successfully inhibited the growth and suppressed the production of lignin degradation enzymes of the pathogen [29], where benzoic acid is identified as the potential inhibitor [29]. Benzoic acid is a plant secondary metabolite that is naturally produced by

plants in response to the biotic and abiotic stresses in the plants [32]. Benzoic acid itself has low toxicity, but there has been concern because of a potential reaction that converts it to benzene. Although benzene is a toxic and carcinogenic compound, the reaction causing this change has a very low chance of occurring in food and environment [33]. Nevertheless, its approval as generally regarded as safe GRAS status, the toxicology of benzoic acid and its derivatives have always been controversial [34]. Owing to its low toxicity, the use of benzoic acid as an antifungal compound could be one of the potential methods to control pathogenic fungi. Therefore, this study aimed to understand and determine the role of benzoic acid on the structural, cellular, and enzymatic activities of *G. boninense* and to evaluate the alteration in wood component structure and patterns of wood degradation caused by *G. boninense* in BSR establishment. The findings could contribute to the development of sustainable control of BSR, aiding in disease-free replantations.

## 2. Materials and Methods

### 2.1. Screening of Benzoic Acid against the Growth of G. boninense Using "Poison Food" Technique

The isolated *Ganoderma boninense* (PER 71) was obtained from the Malaysian Palm Oil Board (MPOB). The culture was maintained by sub-culturing the fungus at regular intervals on potato dextrose agar (PDA) media at 28 °C. Benzoic acid obtained from Chemiz (M) Sdn. Bhd. Posion food technique according to [35,36] was followed. Benzoic acid was added to the heat sterilized media (at 50 °C) to obtain the required concentrations (0, 4.00, 4.25, 4.50, 4.75, 5.00, 5.25, 5.50) mM. The plates were allowed to solidify. The amended media were inoculated with an agar plug (5 mm Ø) of *G. boninense* mycelium excised from an actively growing seven-day-old culture. The PDA plates without benzoic acid served as control. All the plates were incubated at 28 ± 2 °C until the *G. boninense* in control plates were fully colonized. The growth on all plates in diameter was measured and the average value of replicate was taken. The percentage inhibition of diameter growth (PIDG) was measured by using the equation as follows:

$$\text{PIDG (\%)} = \frac{\text{C} - \text{T}}{\text{C}} \times 100 \tag{1}$$

where C is the diameter of the *G. boninense* colony growing in the control plate and T is the diameter of the *G. boninense* colony grown in the treatment plates.

### 2.2. Morphological and Ultrastructural Alterations in the Mycelium of G. boninense

The effect of benzoic acid on the morphological and ultrastructural alterations of *G. boninense* was observed by using scanning electron microscopy (SEM) and high-resolution transmission electron microscopy (HR-TEM), respectively. The observation was carried out for the optimum concentration of benzoic acid, giving the highest percentage inhibition of *G. boninense* growth on the PDA plate.

#### 2.2.1. Scanning Electron Microscopy (SEM)

*G. boninense* mycelium grown on media incorporated with benzoic acid was taken for SEM analysis. Excised samples of 10 mm were fixed in 4% glutaraldehyde for 24 h. The samples were then washed with 0.1 M sodium cacodylate buffer and fixed in osmium tetroxide for 2 h, followed by dehydration in an ethanol series. The samples were dipped in 100% acetone, dried, and coated with gold-palladium before examination with a JEOL Scanning Microscope (JEOL, JSM 6400, Tokyo, Japan).

#### 2.2.2. High-Resolution Transmission Electron Microscopy (HR-TEM)

The mycelium samples were fixed in 6% glutaraldehyde for 24 h at room temperature, rinsed with 0.02 M phosphate buffer, and then subsequently fixed in 2% osmium tetraoxide for 24 h at 20 °C. The samples were dehydrated in HPLC graded ethanol series for five minutes and then dried (SAMDRI 780-B Tousimis, Rockville, MD, USA). The dried mycelia were coated with gold-palladium using a sputter coater (BAL-TEC SDC 050, New York

City, NY, USA). The samples were embedded in resin, followed by polymerization. The ultra-thin section of the mycelial sample (60 nm) were obtained by using a diamond knife (Ultra-cut R, Leica, Washington, DC, USA) and collected in Formvarcoated copper grids (Electron Microscopy Sciences, cat # FCF200, Hatfield, PA, USA). The samples were contrasted in uranyl acetate, lead citrate washed, and examined using a JEM-2100F Field Emission Electron Microscope.

### 2.3. Flow Cytometry

*G. boninense* mycelial plug (5 mm Ø) from 7-day-old culture was inoculated in Potato Dextrose Broth (PDB) incorporated with different concentrations of benzoic acid and incubated for 10 days. PDB with no benzoic acid served as a control. The samples were centrifuged at 9500 rpm for 10 min to obtain the cell (mycelium) before membrane depolarization and propidium iodide influx assay. Results of membrane depolarization and propidium iodide influx assay were analyzed using CytExpert Software (version 2.4.0.28).

### 2.3.1. Membrane Depolarization Assay

The extracted cells were resuspended in 1 mL of phosphate buffered-saline (PBS) pH 7.4 followed by the addition of 5 μg of bis-(1,3-dibutyl barbituric acid) trimethine oxonol [Dibac4(3)]. The samples were analyzed using a flow cytometer (Beckman Coulter Flow Cytometer) [37].

### 2.3.2. Propidium Iodide Influx Assay

The fungal cells were collected by washing with PBS, centrifuged at 9500 rpm for 10 min, and resuspended in PBS. The samples were then incubated for 5 min at room temperature. The uptake of the propidium iodide by the *G. boninense* cells was analyzed using a flow cytometer (Beckman Coulter Flow Cytometer) to identify the fungal membrane permeability [37].

### 2.4. Biodegradation of Oil Palm Wood Blocks

The patterns of oil palm wood degradation by *G. boninense* in the presence of benzoic acid at different concentration was investigated using the method described by Schirp and Wolcott (2005) with slight modifications [38]. The healthy oil palm wood log was cut into 20 mm × 20 mm × 40 mm blocks. The woodblocks were dried in the oven at 60 °C for 48 h, weighed, and labelled individually to obtain the initial dry weight of the woodblock (Wi). The woodblocks were double sterilized for 20 min at 121 °C at 103.4 kPa.

Sterilized tissue culture jars containing 25 mL of PDA media were inoculated with the 5 mm diameter plugs excised from seven-day-old culture of *G. boninense* and incubated at 28 ± 2 °C. The sterilized healthy oil palm woodblocks were dipped in different concentrations of benzoic acid solution (0, 4.00, 4.50, 5.00, and 5.50 mM) for 60 min to ensure the benzoic acid had completely absorbed into the woodblock, followed by air-drying in a laminar flow for 75 min to avoid the leaching of benzoic acid from the woodblock during the incubation. The woodblocks were weighed before and after dipping in benzoic acid solution, and the percentage of benzoic acid solution absorbed was recorded. Benzoic acid treated woodblock was placed individually on the top of the *G. boninense* colonized culture in each jar. All the jars were incubated at 28 °C for 0, 30, 60, 90, and 120 days. The woodblocks without benzoic acid treatments served as control. At the end of each sampling period, five woodblocks were harvested, and the surface of each block was wiped carefully by using a razor blade to remove the mycelium. To obtain the dry weight loss of the woodblock, the woodblocks were dried at 60 °C in the oven until a constant weight was obtained and then weighed (Wf) [39]. The percentage weight loss of wood blocks (WL) for each sampling period was calculated by using an equation:

$$\text{Percentage weight loss (WL)} = \frac{\text{Wi} - \text{Wf}}{\text{Wi}} \times 100 \qquad (2)$$

where Wi is the initial dry weight of the woodblock and Wf is the final dry weight of woodblocks after treatment at each sampling period.

### 2.5. Scanning Electron Microscopy (SEM) Analysis

At the end of each sampling period, the woodblocks from each treatment were harvested, and the mycelia were removed and immediately processed. The transverse sections of the wood blocks (10 mm × 10 mm × 10 mm) were obtained by a razor blade. The samples were then dried, mounted on aluminium stubs, and sputter-coated with gold-palladium (BAL-TEC SCD 005, New York City, NY, USA). The scanning electron microscope (JSM-IT100 InTouch Scope$^{TM}$, JEOL, Tokyo, Japan) was used to determine the structural changes in wood during the degradation period. The sampling was performed at 30-day intervals until 120 days. Healthy wood without any treatment was also examined to compare with degraded wood.

### 2.6. Production of Ligninolytic and Hydrolytic Enzymes by G. boninense

*G. boninense* was cultured in a 250 mL Erlenmeyer flask containing 40 mL of sterilized Potato Dextrose Broth (PDB) and amended with benzoic acid at different concentrations (0, 2.50, 3.50, 4.50, and 5.50 mM). The sterilized rubber wood chips obtained from MPOB (5.0 g) were added to the medium to induce the production of the enzymes. Each of the flasks was inoculated with 5 mm Ø mycelial plugs excised from the actively grown seven-day-old culture of *G. boninense* and allowed to grow at 28 °C under constant shaking (200 ppm) for ten days. The *G. boninense*, along with the wood chip, was centrifuged at 4000 rpm for 40 min at 4 °C. The supernatant was transferred to a fresh tube that served as an enzyme source. The crude was stored at −20 °C for 24 h before enzyme assays.

#### 2.6.1. Lignolytic Enzymes Assay

Three ligninolytic enzymes, namely laccase, lignin peroxidase, and manganese peroxidase, were estimated colorimetrically by UV-Vis spectrometer. The enzyme activity was measured using cell-free crude. All the enzyme activities were expressed in U mL$^{-1}$. All enzyme assays were performed twice with three replicates.

#### 2.6.2. Laccase Enzyme Assay

The laccase activity was measured by the level of oxidation of the nonphenolic, 2, 2′-azino-bis (ABTS) as a substrate at 420 nm (spectrometrically). The assay mixture consists of 0.5 mmol ABTS in 0.1 mol L$^{-1}$ sodium acetate (pH 4.5). The reaction was started by the addition of 100 μL of crude enzyme extract to 2.80 mL of ABTS solution [29,40].

#### 2.6.3. Lignin Peroxidase Enzyme Assay

The lignin peroxidase activity was quantified by an amount of veratryl alcohol oxidized to veratraldehyde. The reaction mixture consisted of 1 mL of 125 mmol sodium tartrate buffer (pH 3.0), 0.50 mL of 10 mmol veratryl alcohol, and 0.50 mL of the crude enzyme extract. The reaction was activated by the addition of 0.50 mL of 2 mmol hydrogen peroxide solution and measured at 310 nm [29].

#### 2.6.4. Manganese Peroxidase Enzyme Assay

The manganese peroxidase activity was measured by the oxidation of guaiacol at 465 nm. The reaction mixture consisted of 0.50 mL each of 100 mmol sodium tartrate (pH 5), 100 mmol guaiacols, 50 mmol hydrogen peroxide, and 0.1 mL of crude enzyme extract [29].

#### 2.6.5. Cellulase, Xylanase, and Amylase Enzymes Assays

Enzyme extracts (250 μL) from each treatment were incubated at 50 °C for 30 min with 250 μL of 1% carboxymethylcellulose (CMC, Fluka AG, Buchs, Switzerland) in 50 mmol sodium citrate buffer (pH 5). Soluble starch and birchwood xylan were used as the substrates instead of CMC for the amylase and xylanase assays, respectively. The reaction was

stopped by the addition of 500 μL of DNS (3,5-dinitrosalicylic acid) to the mixture and was submerged in boiling water (100 °C) for 5 min. The mixture was cooled down to room temperature, and the reactions were monitored at 540 nm using a UV-Vis spectrophotometer. One unit of cellulase and amylase activities was defined as the amount of enzyme required to release 1 μmol of glucose equivalents per minute under given conditions, and one unit of xylanase activity was defined as the amount of enzyme required to release 1 μmol of xylose equivalents per minute under given conditions [39].

### 2.7. Statistical Analysis

The data were analyzed using SAS statistical software (PC-SAS 9.4 system) for normality tests and analysis of variance (ANOVA). The Shapiro-Wilk test was used to check the normality of the data prior to the analysis of variance (ANOVA). The experiment was repeated twice. The means were compared by Tukey's (HSD) test at a 95% probability level.

## 3. Results

### 3.1. Growth of G. boninense on the Media Incorporated with Benzoic Acid

Antifungal activity of benzoic acid at different concentrations against the growth of *G. boninense* was tested by the "poison food" method. The mycelium fully colonized the control plate at 10 days after inoculation, while benzoic acid-incorporated plates showed the slower growth of *G. boninense*. The growth rate of *G. boninense* in control plates was 0.8 cm day$^{-1}$, while in treated plates it was 0.1 to 0.4 cm day$^{-1}$. The percentage inhibition of diameter growth of *G. boninense* on treated and control plates is presented in Table 1.

**Table 1.** Percentage Inhibition of Diameter Growth (PIDG) of *G. boninense* on the media incorporated with different concentrations of benzoic acid.

| Benzoic Acid Concentrations (mM) | Growth Rate (cm Day$^{-1}$) | PIDG (%) |
|:---:|:---:|:---:|
| 0.00 | 0.83 | 0 |
| 4.00 | 0.40 | 62.37 ± 1.4 [e] |
| 4.25 | 0.35 | 68.21 ± 1.7 [d] |
| 4.50 | 0.27 | 69.43 ± 1.5 [d] |
| 4.75 | 0.13 | 75.31 + 1.6 [c] |
| 5.00 | 0.10 | 80.00 + 1.0 [b] |
| 5.25 | 0 | 100 [a] |
| 5.50 | 0 | 100 [a] |

± value represents the standard deviation of the means that were compared by Tukey's (HSD) test at a 95% confidence level. Means with the same letter in a column are not significantly different ($p \leq 0.05$).

The increase in the concentration of benzoic acid resulted in a decrease in the diameter growth of *G. boninense* and increased the percentage inhibition of mycelial growth. There was a significant ($p \leq 0.05$) difference in inhibition of diameter growth in the plates treated with benzoic acid. At the highest tested concentration (5.25 and 5.50 mM), absolute control was detected, where the PIDG was 100% (Table 1). That indicates there was no growth of *G. boninense* in the plates treated with concentrations of over 5.00 mM of benzoic acid.

### 3.2. Alteration in the Ultrastructure of G. boninense

The antifungal activity of benzoic acid that gives the highest inhibition percentage on diameter growth was investigated by analysis of alteration in the anatomical and cellular structure of *G. boninense* mycelia by using a scanning electron microscope (SEM) and High-Resolution Transmission electron microscope (HR-TEM). The untreated sample that served as control shows the healthy, smooth, compact, long, extensive branch and clear separated clamp connection of *G. boninense* mycelia having a diameter of 1.244 μm (Figure 1a). The benzoic acid treatment (5.00 mM) caused severe structural damages to *G. boninense* mycelia. Treated *G. boninense* mycelia were thin, less dense, distorted, and shrivelled, with clear holes and ruptures of 1.435 μm in diameter (Figure 1b).

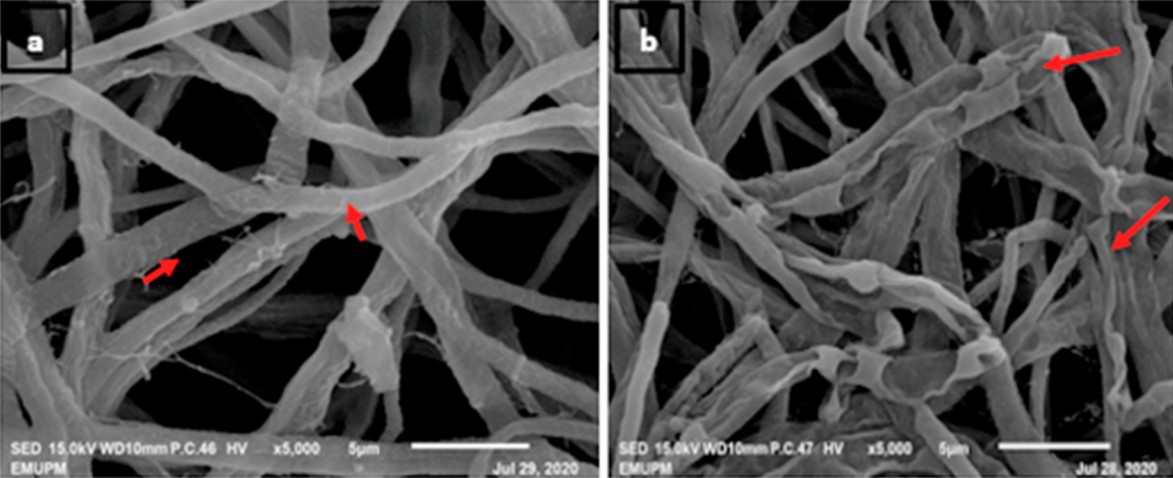

**Figure 1.** Scanning electron micrograph of *G. boninense* mycelia grown on PDA media. (**a**) Smooth, compact structure with distinctive clamp connection (arrow) in the healthy mycelium. (**b**) Mycelium excised from benzoic acid incorporated media showing distortion with shrivelling (arrow), holes and rupture (arrow) in mycelial structure at (5000×).

High-resolution transmission electron microscopic analysis for benzoic acid-treated and healthy mycelia revealed the alteration of the cellular structure of *G. boninense* (Figure 2). Healthy mycelia show the intact cell wall and cell membrane. The Cytoplasm in the mycelium was very dense; however, mitochondria, nucleus, lipid, and Golgi apparatus were prominent (Figure 2a). Benzoic acid treatment caused considerable deformation in mycelia and shrinkage of the cytoplasmic content and loss of organelles. There was clear lysis of the cell wall and the nuclear membrane. The thinning and disruption in the cell membrane, extensive vacuolization, and formation of inclusions at the periphery that may contain glycogen were also observed (Figure 2b).

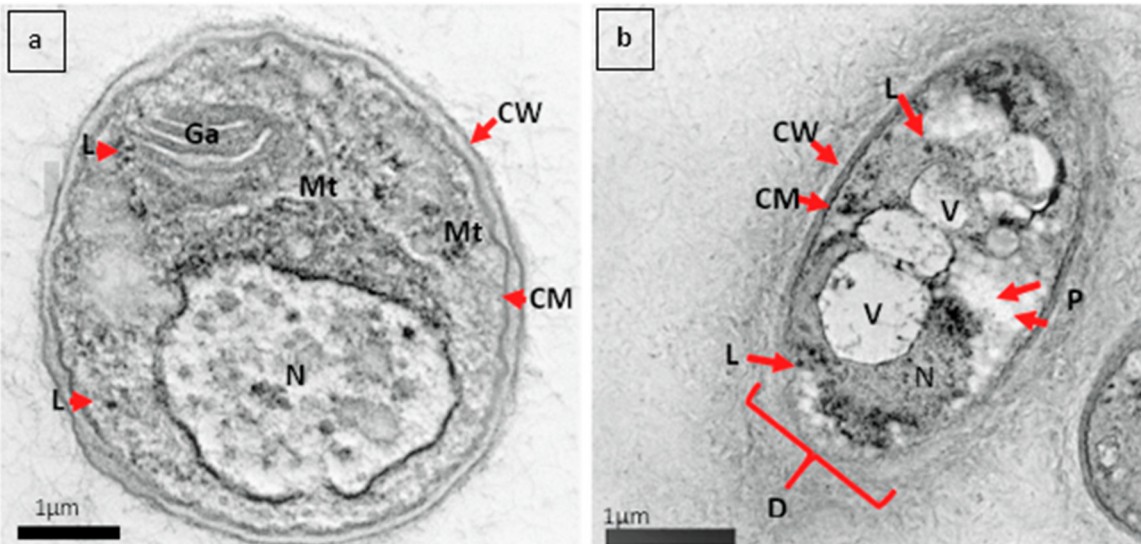

**Figure 2.** Ultra-structure observed via HR-Transmission electron micrograph of *G. boninense* mycelia (**a**) intact cell wall and cell membrane in healthy mycelium (1 μm). (**b**) Mycelium excised from benzoic acid (5 mM) incorporated media showing disintegrated cell wall and cell membrane (arrow), thinning of the cell membrane (arrow), and damage in the nucleus (1 μm). Where, CW: cell wall; CM: cell membrane; Ga: Golgi apparatus; Mt: Mitochondria; N: nucleus; V: vacuole; L: Lipid bodies; P: Polysaccharide inclusions and D: damaged cell wall and cell membrane.

### 3.3. Cell Membrane Integrity and Depolarization

The damage in *G. boninense* cell membrane structure may cause the dissipation of the membrane potential. To investigate the antifungal effect of benzoic acid on the disruption of *G. boninense* cell membrane integrity, the propidium iodide influx assay and membrane depolarization assay was carried out using a flow cytometer. Bis-(1,3-dibutyl barbituric acid) trimethine oxonol [Dibac4(3)] was used to stain membrane damaged by benzoic acid. The stained cell indicates the percentage damage of untreated and treated *G. boninense* cell membrane (Figure 3). T-e least percentage membrane damage (0.24%) was recorded in healthy *G. boninense* cells (Figure 3A) and was significantly low compared to benzoic acid-treated *G. boninense* cells (17.4%) (Figure 3B).

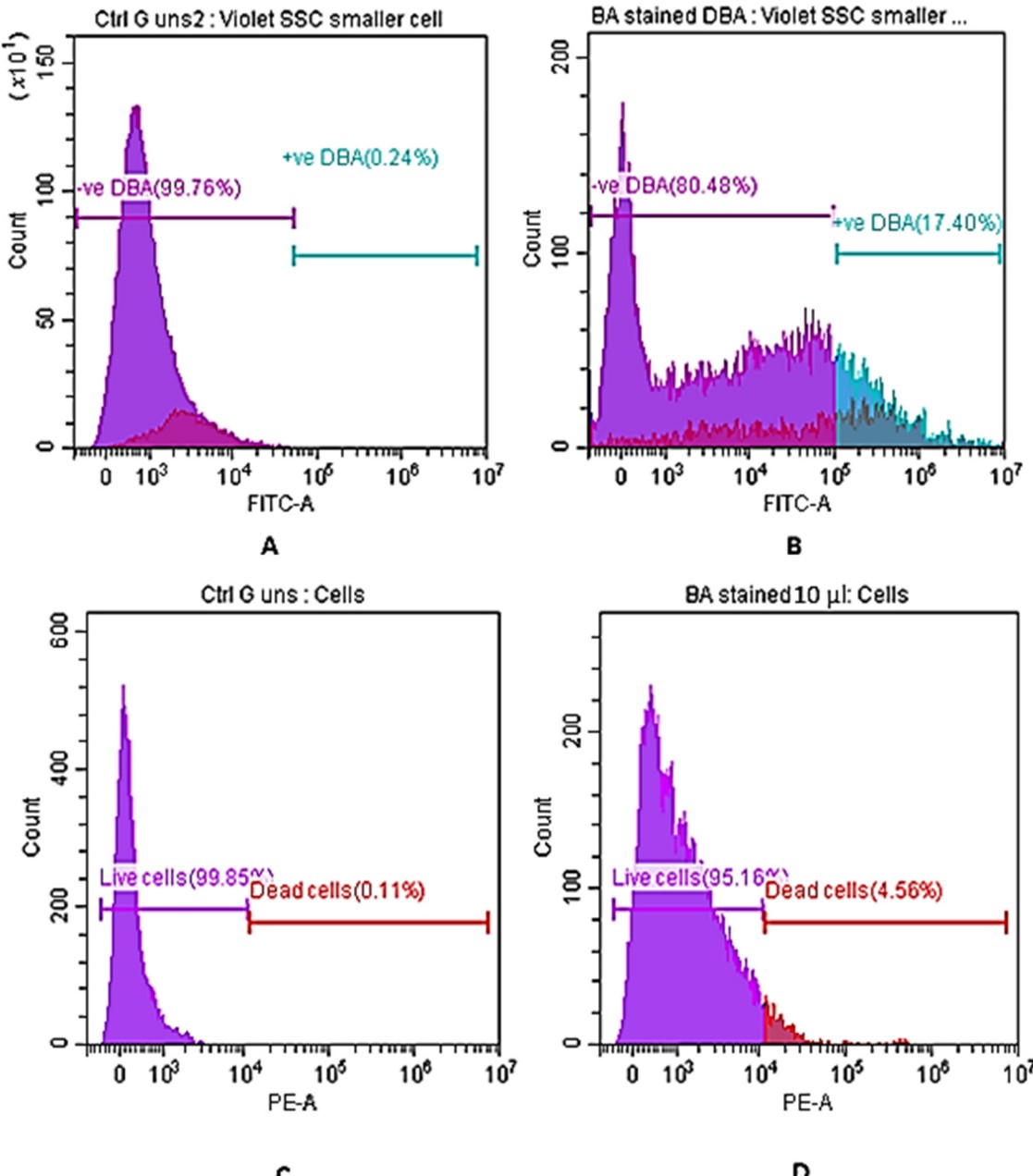

**Figure 3.** The percentage of membrane stained with Dibac4(3) for detection of membrane depolarization of (**A**) untreated, and (**B**) benzoic acid-treated *G. boninense* cell. The percentage of nucleus stained with propidium iodide for detection of living and dead cells of (**C**) untreated and (**D**) benzoic acid-treated *G. boninense* cell based on flow cytometer analyses.

Propidium iodide is a membrane permeability dye that only enters damaged or dead cells. The stained cells from untreated *G. boninense* cell recorded 0.11% dead cells (Figure 3C), whereas *G. boninense* treated with benzoic acid showed 4.56% dead cells (Figure 3D).

### 3.4. Biodegradation of Oil Palm Woodblock

The degradation of oil palm woodblocks by *G. boninense* showed a gradual decrease in the wood dry weight after 30, 60, 90, and 120 days of incubation. The progression of the incubation period increased the colonization of *G. boninense* mycelia and dry weight loss of woodblock. The colonization of mycelia on the surface of benzoic acid-treated woodblock was slow compared to untreated woodblock where it rapidly colonized the wood surface. It took 10 days for the mycelia to cover the woodblocks completely in response to benzoic acid treatment, while in control it took 7 days. Normally, *G. boninense* growing on the media does not show any formation of basidiocarp, whereas *G. boninense* growing in oil palm woodblock showed the formation of basidiocarp within 30 days of incubation. The formation of basidiocarp was observed in all the untreated wood blocks, which increased in size with the increase of incubation time (Figure 4). The benzoic acid-treated woodblock above 4.5 mM did not show any formation of basidiocarp at 120 days of incubation (Figure 4c–e).

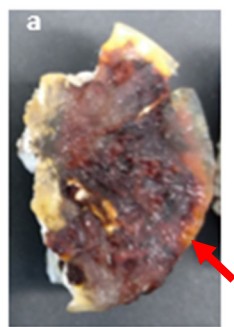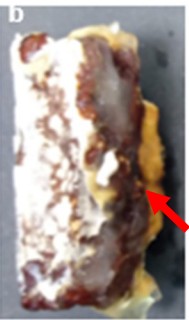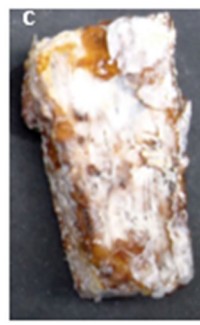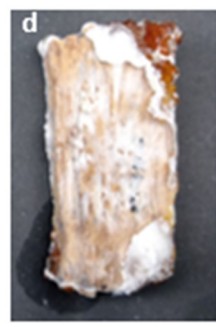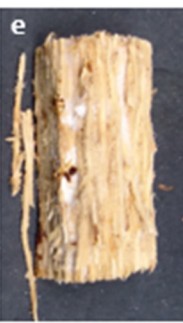

**Figure 4.** *G. boninense* inoculated woodblock after 120 days of inoculation (**a**). Untreated woodblock. Benzoic acid-treated woodblock with different concentrations: (**b**) 4.00 mM, (**c**) 4.50 mM, (**d**) 5.00 mM, and (**e**) 5.50 mM. Arrows indicate the basidiocarp formation on the woodblock.

The loss in wood dry weight at different incubation periods is shown in Figure 5. At the end of the incubation period (120-days), the untreated woodblock showed significantly greatest loss (58.86 ± 5.06) in wood dry weight compared to treated woodblock with different concentrations of benzoic acid. The increase in incubation time caused a significant increase in loss of wood dry weight. On day 30 after incubation, the treated and untreated woodblock showed almost the same weight loss, ranging from 13.36–18.36%, where there was no significant ($p \leq 0.05$) difference. However, at day 60, 90, and 120 after incubation, untreated woodblocks showed a greater loss at 30.76%, 54.51%, and 58.86, respectively, compared to the treated wood blocks. The woodblocks treated with 4.00 mM and 5.00 mM were significantly ($p \leq 0.05$) different in percentage weight loss at the end of the incubation period (120-days). Woodblock treated with the highest concentration of benzoic acid caused a reduction in the dry-weight weight loss of58.82% based on weight loss in untreated woodblock and it is considered as the best concentration in delaying the degradation of oil palm woodblock by *G. boninense*.

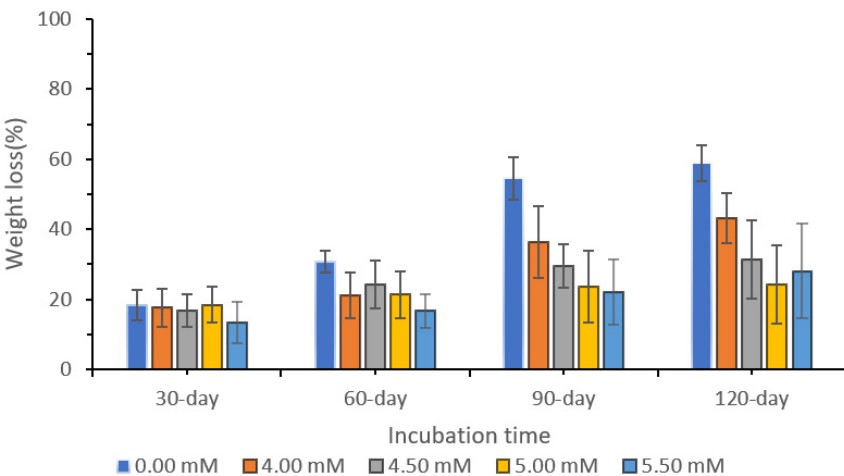

**Figure 5.** Percentage weight loss of untreated and benzoic acid treated oil palm woodblock degraded by *G. boninense* at different incubation times. The error bars represent the standard deviation of the means that were compared by Tukey's (HSD) test at a 95% confidence level.

*3.5. Anatomical Changes in Oil Palm Wood Structure*

　　The healthy oil palm wood structure comprises silica bodies, parenchyma cells, and vascular bundles that made up of fibrous cells and vessels. Scanning electron microscope (SEM) images in Figure 6 show the anatomical changes that occurred in healthy oil palm woodblock after pathogen invasion and benzoic acid treatment at different incubation periods. Observations revealed that the healthy oil palm woodblock has a compact structure consisting of fibres, vessels, parenchymatous cells, and silica bodies. The silica bodies attached to circular craters and were arranged in a pattern of flowers that typifies oil palm wood (Figure 6A,B). The untreated oil palm woodblock revealed the extensive penetration of fungal hyphae through cell fibre and vessels followed by the colonization of hyphae 30 days after the incubation period. Dense mycelia of *G. boninense* were observed on the surface of untreated oil palm woodblocks (Figure 6C).

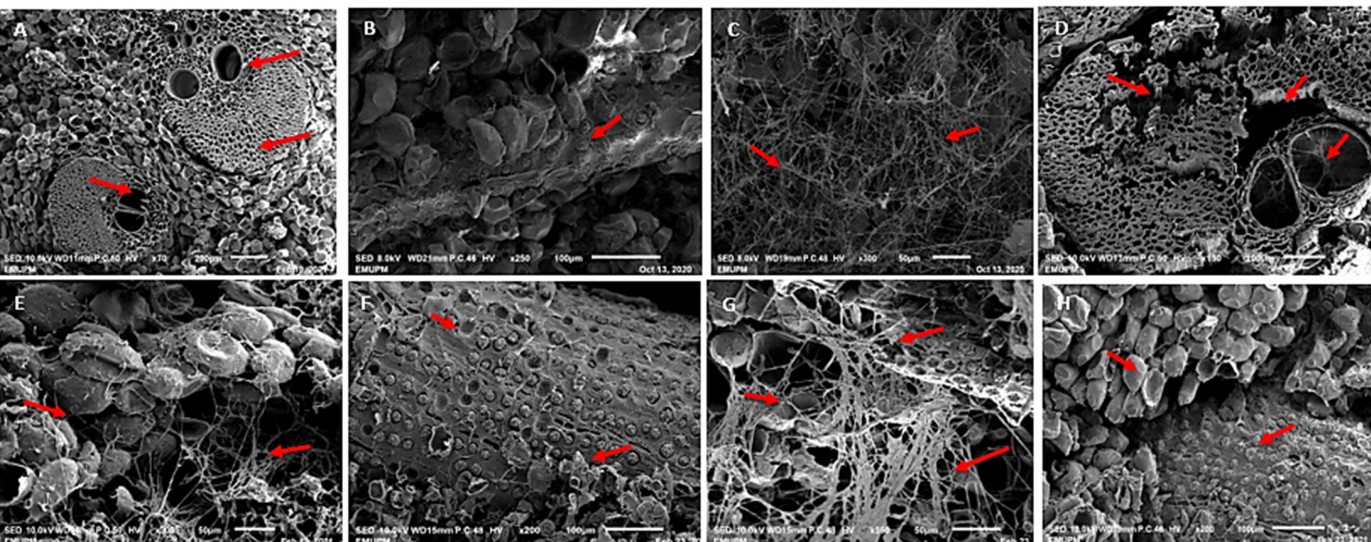

**Figure 6.** Scanning Electron Micrographs of oil palm wood blocks inoculated with *G. boninense*. (**A**) Healthy oil palm wood with compact fibres structure, vascular buddle, and parenchymatous cell (arrow). (**B**) Inoculated healthy wood treated with 5.00 mM benzoic acid, showing intact silica bodies and parenchymatous cell 30 days after inoculation. (**C**) Colonization of *G. boninense* on the surface of untreated wood at 30 days of incubation. (**D**) Inoculated healthy wood treated with 4.00 mM benzoic acid showing separated fibers and hyphae appear thin in structure (arrow) 30 days after inoculation. (**E**) *G. boninense*

colonization on untreated wood shows damage in the parenchymatous cell with boreholes (arrow) and the presence of intense fungal hyphae (arrow) at 60 days of incubation. (**F**) Damage in silica bodies with the presence of fungal hyphae in untreated wood at 90 days of incubation. (**G**) Severe damage in parenchyma cell and silica bodies due to *G. boninense* penetration on untreated wood after 120 days of incubation. (**H**) Inoculated woodblock treated with 5.00 mM benzoic acid, with intact parenchyma cell and scarce presence of hyphae on the silica bodies at 120 days after incubation.

On 60 and 90 days after incubation, further penetration of fungal hyphae occurred and caused damage in the parenchymatous cells. The cells were occupied with fungal hyphae after removal of a lignocellulosic component of the parenchymatous cell, as observed in untreated oil palm woodblock (Figure 6E,F). Treated oil palm woodblock with different concentrations of benzoic acid showed slower progression of penetration by fungal hyphae. There was no visual damage observed on the woodblock structure treated with the highest concentration (5.5 mM) of benzoic acid on day 30 of incubation (Figure 6B), while at the lowest concentration, the treated wood showed the penetration and colonization of fungal hyphae appearing, thin in structure. The penetration of fungal hyphae caused the formation of holes and cavities, and separation of wood fibres (Figure 6D).

Untreated woodblock at day 120 showed severe damage in the structure. The successful penetration and colonization of fungal hyphae resulted in the damage of silica bodies and parenchymatous cells (Figure 6G). Benzoic acid treated oil palm woodblock 120 days after incubation exhibited less colonization of fungal hyphae and apparently less damage in the wood structure. The flower-shaped silica components embedded in the circular craters were intact on the surface of the wood (Figure 6H).

The successful penetration of fungal hyphae and the changes in the wood cells' structure due to fungal attack in control and benzoic acid-treated (4.00 mM) woodblocks indicated that the components of wood, especially lignin, had been degraded. Lignin acts as the first line of defense for the plants. To successfully penetrate the plant cell, lignin needs to be degraded first by enzymatic activities of *G. boninense*.

*3.6. Ligninolytic Enzymes Activity of G. boninense*

Ligninolytic enzymes such as laccase (Lac), lignin peroxidase (LiP), and manganese peroxidase (MnP) produced by *G. boninense* under the influence of benzoic acid at different concentrations are presented (Table 2). Manganese peroxidase production was the highest compared to laccase and lignin peroxidase (MnP > Lac > LiP). Benzoic acid treatment caused the inhibition of laccase and manganese peroxidase production. Laccase production in the benzoic acid-treated medium was 61% lower than in the control medium. Similarly, post benzoic acid-treatment, with increasing concentrations the production of MnP enzyme shows reduced enzyme activity (91%) compared to the control, while an increase in the production of LiP was recorded in different concentrations of benzoic acid-treated *G. boninense* compared to the control.

**Table 2.** Production of ligninolytic enzymes by *G. boninense* under the influence of benzoic acid.

| Benzoic Acid Concentration (mM) | Enzyme Activity | | |
|---|---|---|---|
| | Laccase (U/L) | Lignin Peroxidase (U/mL) | Manganese Peroxidase (U/L) |
| 0.00 | 0.67 [b] | 0.001 [b] | 5.36 [a] |
| 2.50 | 0.33 [c] | 0.066 [a] | 0.49 [e] |
| 3.50 | 0.25 [d] | 0.109 [ba] | 1.57 [b] |
| 4.50 | 1.18 [a] | 0.107 [a] | 1.25 [d] |
| 5.00 | 0.26 [d] | 0.079 [ab] | 1.49 [c] |

The means were compared by Tukey's (HSD) test at a 95% confidence level. Means with the same letter in a column are not significantly different ($p \leq 0.05$).

### 3.7. Hydrolytic Enzymes Activity of G. boninense

The activity of amylase was the highest compared to the xylanase and cellulase activity (amylase > xylanase > cellulose) at $2.04 \pm 0.018$ U mL$^{-1}$. Increasing the concentration of benzoic acid resulted in a reduction in the enzymatic activity of cellulase and xylanase (Table 3). Post benzoic acid-treatment, a significant decrease (86%) in the production of cellulase in comparison with control was recorded, whereas post benzoic-acid treatment, a significant increase (10%) in the production of amylase was recorded when compared to the control. The production of xylanase at the highest concentration was decreased by 66%.

**Table 3.** Production of hydrolytic enzymes by *G. boninense* under the influence of benzoic acid.

| Benzoic Acid Concentration (mM) | Enzymes Activity (U/mL) | | |
| --- | --- | --- | --- |
| | Cellulase | Amylase | Xylanase |
| 0.00 | 0.64 [a] | 2.04 [a] | 1.45 [a] |
| 2.50 | 0.19 [c] | 2.24 [a] | 1.16 [b] |
| 3.50 | 0.09 [e] | 2.02 [a] | 1.51 [a] |
| 4.50 | 0.10 [d] | 1.92 [a] | 1.03 [b] |
| 5.00 | 0.23 [b] | 2.10 [a] | 0.50 [c] |

The means were compared by Tukey's (HSD) test at a 95% confidence level. Means with the same letter in a column are not significantly different ($p \leq 0.05$).

## 4. Discussion

Benzoic acid is a plant secondary metabolite, and it is a key component in the phenylpropanoid pathway that regulates the defense system in plants against biotic and abiotic conditions [32]. The use of plant secondary metabolites (phenolic compounds) in controlling plant pathogenic fungi has become a new trend in current research studies [35,41]. It produces various phytochemicals with various functions such as pigmentation, growth enhancement, and protection from pathogens [32]. These possess antimicrobial properties against various microorganisms such as fungi, bacteria, and viruses [32]. Various studies have been conducted to control diseases caused by pathogenic fungi such as *Aspergillus parasiticus* [41], *Fusarium verticillioides* [42], *Botrytis cinerea* [43], and *Ganoderma boninense* [35] using phenolic compounds.

Benzoic acid efficiently inhibited mycelial growth on PDA. The inhibition was concentration dependent. The findings are in line with those who reported that benzoic acid at 5 mM concentration had the greatest inhibitory effect on *G. boninense* in in vitro tests [35]. Elsewhere, treatment with 1 mM benzoic acid delayed and hindered the mycelial growth towards *Eutypa lata* [44]. Similarly, a study reported the antagonistic effect of different phenolic compounds against the growth of *G. boninense* and enzyme production. Amongst all tested phenolic compounds, benzoic acid was the greatest inhibitor of the growth and production of enzymes by *G. boninense* [35]. The mechanism of benzoic acid in reducing the infection caused by *G. boninense* is related to the production of defense enzymes and lignin content by the host [29]. It was suggested that changing the lignin-degrading activity of *Ganoderma* could be a possible approach to alleviate the spread of *Ganoderma* in the future [45]. In general, phenolic compounds inhibit fungal growth by reacting with proteins and causing a loss of enzymatic function [46].

Further observations using SEM and HR-TEM suggested the mechanism of inhibition of *G. boninense* was related to the alteration of the morphological and ultracellular structure of mycelia. The antifungal effect of benzoic acid leads to a loss in mycelial membrane integrity and cell membrane polarity. The integrity of the plasma membrane played a crucial role in maintaining fungal viability and membrane damage could lead to loss of intracellular components [47]. Disruption of the fungal anatomical and cellular structure indicates the loss of membrane permeability and rigidity, causing damage in the cell membrane of fungi, supported by HR-TEM analysis.

The effect of benzoic acid interfering with the cell membrane of *G. boninense* was evaluated by membrane depolarization and propidium iodide (PI) influx assays. Membrane

depolarization and PI influx assays are useful to measure the loss of membrane integrity [37] where Dibac4(3) and propidium iodide, respectively, are used as dye that can only enter the depolarized cells. The results showed an increased fluorescence of dye in benzoic acid-treated *G. boninense* cells, indicating swift breakdown of the cell membrane of *G. Boninense*, which affected the cell viability. These findings are in line with those who reported the effect of curcumin as polyphenolic compounds on *Candida albicans*, where the membrane depolarization and propidium iodide assays showed an increased fluorescence of Dibac4(3) as it exhibits high voltage sensitivity and enters the damaged membrane [37]. It was suggested that the antifungal curcumin dissipates the cytoplasmic membrane potential of *C. albicans* [37]. The flow cytometry analysis confirmed that benzoic acid influences the normal functioning of the *G. boninense* membrane by leading to fungal inhibition and cell death [48]. Benzoic acid exhibited a potent antifungal activity via a mechanism associated with membrane disruption as indicated by membrane permeabilization and depolarization. It is a primary target of antifungals that can lead to death in fungus [49,50]. The cell membrane of fungi is made of a lipid layer that maintains the integrity of the cell membrane [51]. This layer was modified by fungicide of the aromatic hydrocarbon group, impacting the functionality of the microbial membrane system. The cell membrane of treated fungi becomes sensitive to solar radiation, which then destroys the structure of linoleic acid, a common membrane lipid [52] leading to cell death.

The role of benzoic acid in maintaining the wood structure and delaying the penetration of fungal hyphae and the role of *G. boninense* in the degradation of oil palm woodblock was studied. The infection initiated after the exposure of woodblock to the fungal mycelia allowing penetration to occur at the early stage of infection. After 30 days of incubation, the penetration of fungal hyphae resulted in the development of holes on parenchymatous tissues and separation of wood fibers, indicating that delignification had occurred. There was the least damage on the wood structure at the initial stage of infection due to the presence of lignin as the first barrier of defense in the wood. The removal of lignin at the initial stage of infection makes the plant lose its defense in protecting the polysaccharide components of the wood, further leading to the decomposition of cellulose as white residue [53]. However, an increase in the incubation time resulted in further penetration and colonization of fungal hyphae indicating the lignocellulosic materials including lignin and other polysaccharides components of woods have been degraded. Lignin, cellulose, and hemicellulose are the major wood lignocellulosic material that provides protection and mechanical support to the weight of the plants, it represents 20–30%, 50%, and 25–30% of the total wood dry weight, respectively [54,55]. The degradation of these components resulted in the significant loss of wood dry weight four months after inoculation in untreated woodblock, as evident in this study. The findings are in line with those who reported significant weight loss in woodblocks post-*Ganoderma* infection [39].

A similar study was also reported by Rees, where the primary infection starts when oil palm roots in contact with fungal mycelia, and penetration occurred through the unwounded root. Invasion of root cortex and parenchyma resulted in the development of holes in all cell wall layers and rapid progression of colonization of the lower stem [27] increasing wood porosity and easier penetration [23]. Degradation of cell wall resulted in the formation of cavities within cell wall [27], separation of the parenchymatous cells and fibers in the early stage. In the advanced stages of degradation, all cell types showed the formation of erosion channels and boreholes causing a loss in the rigidity of the cell, then eventually cell collapse [56]. The appearance of wood affected by an alteration in wood structure indicates the changes in mechanical and physical properties as well as chemical components of wood [22]. The changes in the physical properties of the wood can be observed by a reduction in wood weight due to the degradation of lignocellulosic materials [22].

The delayed infection was observed in benzoic acid-treated woodblocks. The ability of *G. boninense* in wood degradation is based on its specialized structure and enzyme production to neutralize plants defense system. It is essential for the successful penetration

of fungal hyphae and further degradation of lignocellulosic materials of oil palm wood. In the presence of benzoic acid in the woodblock, the fungal pathogen might need to spend more energy by producing the ligninolytic enzymes to overcome the defense system induced by the benzoic acid in the wood. Therefore, at the early stage of infection, no delignification occurred in benzoic acid-treated woodblock. However, at the advanced stage of infection fungal hyphae have penetrated the wood, showing the least damage in the wood structure.

The role of benzoic acid in maintaining the wood structure and prevent the penetration of fungal hyphae is associated with the production of lignin and strengthening the first line of the defense system in the plants. That can be observed from the wood structure where the treated woodblock with benzoic acid showed less delignification and lignocellulosic materials remain compact in comparison with untreated woodblock. Benzoic acid is a core phenylpropanoid pathway. It is associated with the production of lignin and regulates the defense system of plants against biotic and abiotic condition [32]. Benzoic acid performs as a precursor for the various primary and secondary metabolites in plants. One of the plant hormones derived from benzoic acid is salicylic acid, which is one of the key endogenous signals involved in defense response activation [57]. Synthesis of salicylic acid through phenylalanine pathway converts benzoic acid to salicylic acid via benzoic acid 2-hydroxylase [58]. The accumulation of salicylic acid in response to the penetration of pathogens creates the local defense and induces resistance via localized plant cell death at the initial attachment of the pathogen in the infection site [59]. The inducing of plant defense response at the early infection can reduce the penetration and spread of pathogens via the infection site [59]. Lignin is one of the major components of oil palm wood fibre. It contributes to the compression strength of the woody stem [60]. The degradation of lignin is considered the rate-limiting step in the infection process [29]. The role of lignocellulose turnover is key, since lignin is the most recalcitrant component of wood [30]. Therefore, the level of wood resistance to degradation depends on the lignin composition in the cell wall. Strengthening the lignin component and increased activity of defense enzymes could be one of the potential ways to stop the infection by *G. boninense*.

*G. boninense* produces ligninolytic and hydrolytic enzymes to overcome the lignin barrier of the plants [29]. These enzymes are responsible for lignin modification and degradation of wood polysaccharides [61]. Lignin degradation is mainly accomplished by the activity of laccase (Lac), manganese peroxidase (MnP), and lignin peroxidase (LiP) that are the major group of ligninolytic enzymes produced by wood degrading fungi [62]. We found that the production of MnP was 5-folds higher compared to Lac and LiP, and a significant reduction in the production of MnP was identified in the presence of benzoic acid. Production of these peroxidase enzymes (MnP and LiP) during degradation at the advanced stages, contribute to the lignin degradation process that was taking place in the diseased or untreated healthy blocks [30,39]. The production of LiP was higher in treated samples with benzoic acid compared to control. An increase in the production of LiP could be due to benzoic acid involed in the protection of lignin in the cell wall. Lignolytic enzymes produce to degrade the lignin in the cell, enabling the fungus to access the cellulose components and leading to the awakening of plant defense response, resulting in reducing lignin components in the cell wall [63].

Once the lignin is successfully degraded by the activity of ligninolytic enzymes, *G. boninense* produces hydrolytic enzymes such as cellulase, amylase, and xylanase that play an important role in the degradation of polysaccharides components of the wood. During the degradation process, carbohydrates are broken down into simple sugar by fungal enzymes [64]. The rapid and extensive degradation of wood components by enzymatic activities of the pathogen is characterized by wood bleaching due to lignin removal [29]. The hydrolytic enzymes are important in the degradation of cellulose and hemicellulose into simple molecules that can be utilized by the pathogen for nourishment [28]. The production of hydrolytic enzymes was significantly low compared to the ligninolytic enzymes observed in our study. This could be due to ligninolytic enzyme production in

deconstructing lignin and overcoming the first line of defense to penetrate the plant cell. The hydrolytic enzyme will only produce when the pathogen successfully penetrates the plant cell for nourishment after the loss of energy during penetration, which is not the case in benzoic acid treated wood at high concentration. We found that the production of amylase is higher compared to other hydrolytic enzymes. Amylase is the most effective starch degrader and caused the greatest losses in weight in oil palm block [39]. Higher activity of amylase enzymes might be related to the high content of starch and sugar in oil palm wood intended for biodegradation of oil palm wood [39,65]. The results of this study verify that benzoic at 5 mM to be effective in suppressing *G. boninense* and delay the pathogen establishment in woodblock. It could be an effective controller for the BSR disease and hence can be considered as a potential compound to treat the BSR infected stumps and debris reducing the inoculum pressure for future replanting.

## 5. Conclusions

The present work provides knowledge on the mechanism of benzoic acid on suppression of *G. boninense* via alteration in mycelial morphology, cellular ultrastructure, and enzymatic activities. Further, the study demonstrates the role of benzoic acid in maintaining plant cell wall structure and delay in pathogen establishment in the woodblocks. *G. boninense* infection causes extensive degradation of oil palm wood, resulting in a significant loss in the wood dry weight after four months of the incubation period. This determines the ability of *G. boninense* to simultaneously or selectively degrade the major components of oil palm wood that give a major contribution to the wood dry weight. Oil palm woodblock treated with the highest concentration of benzoic acid showed significant inhibition of ligninolytic and hydrolytic enzymes as well as maintenance of the wood components, indicated by less decrement in dry weight of oil palm woodblock and reduced anatomical damage compared to untreated wood. The results confirmed the use of benzoic acid could prevent degradation of oil palm wood caused by *G. boninense* infection, leading to sustainable management or delaying the BSR disease establishment. However, field trials and further detailed studies on the degradation of chemical components of oil palm wood are required to confirm the mechanism of *G. boninense* in wood degradation and to prove the consistency in current findings which may contribute to reduced land expansion to create new disease-free land for oil palm planting.

**Author Contributions:** R.F. and D.G. conducted the experiment, R.F. analysed the data and wrote the manuscript, Y.S. conceptualized, secured grant, visualized, corrected and review the manuscript, Y.S. and K.A. supervised, coordinated and revised the research flow, A.S. review the manuscript. All authors have read and agreed to the published version of the manuscript.

**Funding:** This research was funded by Ministry of Education, Malaysia under the Fundamental Research Grant Scheme (FRGS/1/2019/STG05/UPM/02/22).

**Acknowledgments:** The authors would like to express sincere thanks to the Ministry of Higher Education (MOHE) Malaysia for providing financial support under the project under Fundamental Research Grant Scheme (FRGS/1/2019/STG05/UPM02/22) and Universiti Putra Malaysia.

**Conflicts of Interest:** The authors declare no conflict of interest.

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
