# Peer review of "Suppression of Ganoderma boninense Using Benzoic Acid: Impact on Cellular Ultrastructure and Anatomical Changes in Oil Palm Wood"

_forests, doi:10.3390/f12091231_

Round 1
Reviewer 1 Report
Title-
“using” should not be italicized
Introduction-
Some English edits in the introduction need to be addresses, especially in the second paragraph. Line 67: should read fruit bunches, and ... were dead or collapsed...
Line 68: presence of basidiocarps
Line 69: show that the fungus has...
Line 71-Line73 need to be reworded. Example: Ganoderma boninense is a white-rot fungus due to its ability to enzymatically degrade lignin, a component of oil palm wood. This enzymatic degradation causes a decrease in wood strength during the initial stages of decay.
Line 76-Line 78: Simultaneous degradation breaks down all structural components, lignin, cellulose and hemicellulose, whereas, selective degradation breaks down lignin and hemicellulose leaving cellulose-rich wood.
Please define LiP and MnP. Not all readers know that corresponds to lignin peroxidase and manganese peroxidase. It would also help to discuss a bit about the mode of action of these 2 enzymes. As well as laccase.
Some background on benzoic acid would be beneficial in the intro as well. Has this been used as a deterrent before? What are the ecological/environmental concerns using benzoic acid as a treatment for Ganoderma?
M & M-
Line 132 – phosphate buffers? Multiple buffers? If not, remove the s on buffers.
The authors could condense several sections of the M & M. Example: section 2.2 can be added to section 2.1. Then relabel section 2.2 Imaging with subheaders 2.2.1 SEM 2.2.2 HR-TEM. Section 2.5 can be condensed to the 2.2.1 SEM section.
Section 2.4 –
Did the authors have to calculate for moisture loss in the culture jars? Was water added at regular intervals?
Was there a standardized method you followed for this test? Is that the Schirp and Wolcott method? Why not use an actual standardized method for wood blocks not composite wood? Something like EN113? Or AWPA E10?
What was the need for the double sterilization?
How long and what temp were the blocks dried for before calculating % wt loss? Please add this information.
Section 2.6, Did the authors calculate total protein (Bradford or Lowry Assay) in the supernatant and crude extracts? This needs to be done to adequately quantify laccase, LiP and MnP in these fractions. Same for the hydrolytic enzymes tested too. This is vital information.
Did the authors follow protocols for the enzyme assays? There is no citation for the laccase assay.
Results –
Section 3.1. What is the “food poison method”. This needs a citation.
Section 3.4 The decay numbers are not surprising. Considering all treatments showed at least 30% wt loss by 120 days doesn’t really indicate a decent treatment. Did the authors consider vacuum treating the blocks with benzoic acid? What is the likelihood that benzoic acid leached out of the blocks into the agar in the test jars? Any leaching tests conducted?
Where are the statistical values on Figure 5? Was any treatment significantly different? This must be added.
Line 339 the authors used both present tense and past tense. Please adjust to one or the other not both.
Section 3.7 reads “increasing the concentration of benzoic acid resulted in a reduction in the enzymatic activities of cellulase, amylase and xylanase”. Amylase was not reduced ... statistical analysis says so. Please amend this statement.
Discussion –
Line 418 reads “became a new trend in current research studies” please cite these studies.
Line 425 needs to be rewritten and cited
Line 435, Line 479, Line 480 defense is misspelled
Line 440, Line 455, Line 456 uncapitalize boninense
Line 447-Line 448 doesn’t make sense. What is an essential component of fungal cell membranes? Lipids?
Line 466 change fungus to fungi
Line 474-Line 475 the authors used both present tense and past tense. Please adjust to one or the other not both.
Line 515-Line 519 make this into 2 sentences
Conclusions -
Needs minor english edits.
Author Response
Dear Reviewer
Thank you very much for the careful and thorough reading of this manuscript and for the thoughtful comments and constructive suggestions, which helped us to improve the quality of this manuscript. Our response is attached and the corrections are highlighted (yellow) in the main document.

Reviewer 2 Report
The manuscript titled “ Suppression of Ganoderma boninense using Benzoic Acid: Impact on Ultrastructure of Pathogen and Anatomical Changes in Oil Palm Wood“. I find the idea interesting and in line with the aim of the journal. I have some concerns about the experimental setup to justify what the authors claim. Moreover, the rationale behind some of the data presented was not entirely clear. I also recommend to the authors to improve their references by conducting a more extensive review on international literature. Particularly, in the introduction statements are not supported by the references selected by the authors. The logic of some sentences is also questionable. Below is my point-to-point analysis of the manuscript.
- I suggest modifying the title should be more crisp and brief.
- Abstract introductory statement is too long, it has to be improved with the more specific rationale of the study. The abstract should have crisp information about aim materials method result and conclusion, which I don't find in the present form of an abstract.
- My main concern of a manuscript is the statical test as in the above statement author took 3 replication. what is the value of n while calculating ANOVA? Author mention n= 3 n Value (4) used in the manuscript is too few to examine the normal distribution of variables in the sample, however, the Shapiro-Wilk test is appropriate for samples from 3 to 5000 but for the lesser value of n, it receives the non-normal distribution. Thus ANOVA that is a parametrical test is incorrect for such small samples.
- What method was used to check Normality?
- Figure 5 in the manuscript is with error bars, author should mention what parameter was taken, standard error, standard deviation or it is showing 95% confidence interval.
Author Response
Dear Reviewer
We would like to thank you for the careful and thorough reading of this manuscript and for the thoughtful comments and constructive suggestions, which helped us to improve the quality of this manuscript. Our response is attached and the corrections are highlighted (yellow) in the main document.
Regards

Round 2
Reviewer 1 Report
Manuscript looks great with the added updates and changes. I do think future experiments should measure protein concentration; however.
Can the authors calculate retention of benzoic acid for the wood blocks? This calculation can be found in AWPA Standard E10. This is the most appropriate way to report treatment conditions.
Reviewer 2 Report
The author has modified the manuscript as per my suggestion.